# HACMan: Learning Hybrid Actor-Critic Maps for 6D Non-Prehensile Manipulation

**Wenxuan Zhou**[1,2], **Bowen Jiang**[1], **Fan Yang**[1], **Chris Paxton**[2*], **David Held**[1*]

**Abstract:** Manipulating objects without grasping them is an essential component of human dexterity, referred to as non-prehensile manipulation. Non-prehensile manipulation may enable more complex interactions with the objects, but also presents challenges in reasoning about gripper-object interactions. In this work, we introduce Hybrid Actor-Critic Maps for Manipulation (HACMan), a reinforcement learning approach for 6D non-prehensile manipulation of objects using point cloud observations. HACMan proposes a *temporally-abstracted* and *spatially-grounded* object-centric action representation that consists of selecting a contact location from the object point cloud and a set of motion parameters describing how the robot will move after making contact. We modify an existing off-policy RL algorithm to learn in this hybrid discrete-continuous action representation. We evaluate HACMan on a 6D object pose alignment task in both simulation and in the real world. On the hardest version of our task, with randomized initial poses, randomized 6D goals, and diverse object categories, our policy demonstrates strong generalization to unseen object categories without a performance drop, achieving an 89% success rate on unseen objects in simulation and 50% success rate with zero-shot transfer in the real world. Compared to alternative action representations, HACMan achieves a success rate more than three times higher than the best baseline. With zero-shot sim2real transfer, our policy can successfully manipulate unseen objects in the real world for challenging non-planar goals, using dynamic and contact-rich non-prehensile skills. Videos can be found on the project website: https://hacman-2023.github.io.

**Keywords:** Action Representation, Reinforcement Learning with 3D Vision, Non-prehensile Manipulation

## 1 Introduction

The ability to manipulate objects in ways beyond grasping is a critical aspect of human dexterity. Non-prehensile manipulation, such as pushing, flipping, toppling, and sliding objects, is essential for a wide variety of tasks where objects are difficult to grasp or where workspaces are cluttered or confined. However, non-prehensile manipulation remains challenging for robots; previous work has only shown results with limited object generalization [1, 2] or limited motion complexity, such as planar pushing or manipulating articulated objects with limited degrees of freedom [3, 4, 5]. We propose a method that generalizes across object geometries while showing versatile interactions for complex non-prehensile manipulation tasks.

We present **H**ybrid **A**ctor-**C**ritic Maps for **Man**ipulation (HACMan), a reinforcement learning (RL) approach for non-prehensile manipulation from point cloud observations. The first technical contribution of HACMan is to propose an **object-centric action representation** that is **temporally-abstracted** and **spatially-grounded**. The agent selects a contact location and a set of motion parameters determining the trajectory it should take after making contact. The contact location is selected from the observed object point cloud which provides spatial grounding. At the same time, the robot decisions become more temporally-abstracted because we focus on only learning the contact-rich portions of the action.

The second technical contribution of HACMan is to incorporate the proposed action representation in an actor-critic RL framework. Since the contact location is defined over a discrete action space (selecting a contact point among the points in the object point cloud) and the motion parameters

---

[1]Robotics Institute, Carnegie Mellon University [2]Meta AI [*]Equal Advising

7th Conference on Robot Learning (CoRL 2023), Atlanta, USA.

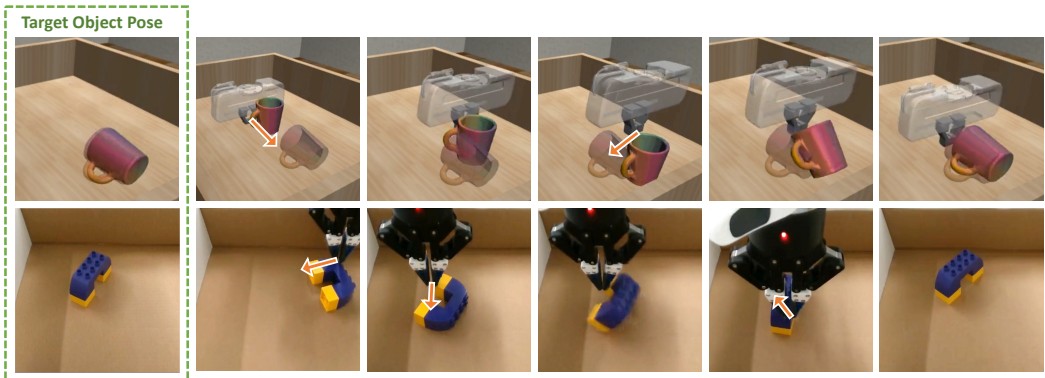

Figure 1: We propose HACMan (**H**ybrid **A**ctor-**C**ritic **Ma**ps for **Man**ipulation), which allows non-prehensile manipulation of unseen objects into arbitrary stable poses. With HACMan, the robot learns to push, tilt, and flip the object to reach the target pose, which is shown in the first column and in the top row with transparency. The policy allows for dynamic object motions with complex contact events in both simulation (top) and in the real world (bottom). The performance of the policy is best understood from the videos on the website: https://hacman-2023.github.io.

(defining the trajectory after contact) are defined over a continuous action space, our action representation is in a hybrid discrete-continuous action space. In HACMan, the actor network outputs *per-point* continuous motion parameters and the critic network predicts *per-point* Q-values over the object point cloud. Different from common continuous action space RL algorithms [6, 7], the per-point Q-values are used both to update the actor and also to compute the probability for selecting the contact location. We modify the update rule of an existing off-policy RL algorithm to incorporate such a hybrid action space.

We apply HACMan to a 6D object pose alignment task with randomized initial object poses, randomized 6D goal poses, and diverse object geometries (Fig. 1). In simulation, our policy generalizes to unseen objects without a performance drop, obtaining an 89% success rate on unseen objects. In addition, HACMan achieves a training success rate more than three times higher than the best baseline with an alternative action representation. We also perform real robot experiments with zero-shot sim2real transfer, in which the learned policy performs dynamic object interactions over unseen objects of diverse shapes with non-planar goals. Our contributions include:

- We propose a novel object-centric action representation based on 3D spatial action maps to learn complex non-prehensile interactions. We also modify an existing off-policy RL algorithm to incorporate such a hybrid discrete-continuous action space.

- The proposed action representation demonstrates substantive improvements of performance over the baselines and shows strong generalization to unseen objects.

- The learned policy showcases complex contact-rich and dynamic manipulation skills, including pushing, tilting, and flipping, shown both in simulation and with a real robot.

## 2 Related Work

**Non-prehensile manipulation.** Non-prehensile manipulation is defined as manipulating objects without grasping them [8]. Many non-prehensile manipulation tasks involve complex contact events among the robot, the object, and the environment, which lead to significant challenges in state-estimation, planning and control [1, 2, 9, 5]. Recent work has applied learning-based methods in non-prehensile manipulation, but they are limited in terms of either skill complexity [10, 3, 4, 11] or object generalization [12, 11]. In contrast, our work shows 6D object manipulation involving more complex object interactions while also generalizing to a large variety of unseen object geometries.

**Visual Reinforcement Learning with Point Clouds.** Recent research has explored various ways of incorporating point clouds into RL [13, 14]. To overcome the optimization difficulties, previous work has tried pre-training the feature extractor with an auxiliary loss [15], initializing the RL policy with behavior cloning [16], or using student-teacher training [17, 18] (see detailed discussion in Appendix F). Our method does not require these additional training procedures due to the benefits of the proposed action representation. In the experiments, we show that the baselines following the most relevant previous work [13, 17, 18] struggles when the task becomes more complex.

**Spatial action maps.** Similar to our method, recent work has explored spatial action maps that are densely coupled with visual input instead of compressing it into a global embedding, based on images [19, 4, 20], point clouds [21, 3, 22], or voxels [23]. Unlike previous works with spatial action maps that consider one-shot decision making (similar to a bandit problem) [3, 4, 10] or rely on expert demonstrations with ground truth actions [19, 21, 23], our method reasons over multi-step sequences with no expert demonstrations. For example, Xu et al. [4] only chooses a single contact location followed by an action sequence, rather than a sequence of contact interactions. Unlike previous work in spatial action maps that uses DQN with discrete actions [24, 25, 10, 20], our hybrid discrete-continuous action space allows the robot to perform actions without discretization. Furthermore, we demonstrate the benefit of spatial action representations when applied to a 6D non-prehensile manipulation task, which is more challenging than the pick-and-place and articulated object manipulation tasks in previous work.

**RL with hybrid discrete-continuous action spaces.** Most RL algorithms focus on either a discrete action space [26] or a continuous action space [7, 6, 27]. However, certain applications are defined over a hybrid action space where the agent selects a discrete action and a continuous parameter for the action [28, 29, 30]. Unlike previous work, our hybrid action space uses a spatial action representation in which the discrete actions are defined over a map of visual inputs. The closest to our work is Feldman et al. [20] in terms of applying RL to spatial action maps, but they only consider a finite horizon of 2. We include formal definitions of the policies over the hybrid action space and modify the loss functions and exploration accordingly.

## 3   Preliminaries

A Markov Decision Process (MDP) models a sequential stochastic process. An MDP can be represented as a tuple $(S, A, P, r)$, where $S$ is the state space; $A$ is the action space; $P(s_{t+1}|s_t, a_t)$ is the transition probability function, which models the probability of reaching state $s_{t+1}$ from state $s_t$ by taking action $a_t$; $r(s_t, a_t, s_{t+1})$ is the immediate reward at time $t$. The objective is to maximize the return $R_t$, defined as the cumulative discounted reward $R_t = \sum_{i=0}^{\infty} \gamma^i r_{t+i}$. Given a policy $\pi$, the Q-function is defined as $Q^\pi(s, a) = \mathbb{E}_\pi[R_t|s_t = s, a_t = a]$.

HACMan is built on top of Q-learning-based off-policy algorithms with a continuous action space [6, 27, 29]. In these algorithms, we define a deterministic policy $\pi_\theta$ parameterized by $\theta$ and a Q-function $Q_\phi$ parameterized by $\phi$. Note that since the policy is deterministic, we use epsilon-greedy during exploration. Given a dataset $D$ with transitions $(s_t, a_t, s_{t+1})$, the Q-function loss is defined according to the Bellman residual:

$$L(\phi) = \mathbb{E}_{s_t, a_t, s_{t+1} \sim D}[(Q_\phi(s_t, a_t) - y_t)^2], \tag{1}$$

where $y_t$ is defined as:

$$y_t = r_t + \gamma Q_\phi(s_{t+1}, \pi_\theta(s_{t+1})). \tag{2}$$

The policy loss for $\pi_\theta$ is defined to maximize the Q-function:

$$J(\theta) = -\mathbb{E}_{s_t \sim D}[Q^{\pi_\theta}(s_t, a_t)|_{a_t = \pi_\theta(s_t)}]. \tag{3}$$

## 4   Problem Statement and Assumptions

We focus on the task of 6D object pose alignment with non-prehensile manipulation. The objective of the robot is to perform a sequence of non-prehensile actions (i.e. pushing, flipping) to move an object on the table into a target goal pose. We assume that the goals are stable object poses on the table. The robot policy observes the point cloud of the scene from depth cameras, denoted as $\mathcal{X}$. We assume that the point cloud observation is segmented between the background and the object to be manipulated. Thus, the full point cloud $\mathcal{X}$ consists of the object point cloud $\mathcal{X}^{obj}$ and the background point cloud $\mathcal{X}^b$. The feature for each point is a 4-dimensional vector, including a 1-dimensional segmentation mask and a 3-dimensional goal flow vector (will be defined in Section 5.3).

## 5   Method

### 5.1   Action Representation

We propose an object-centric action space that consists of two parts: a **contact location** $a^{loc}$ on an object and **motion parameters** $a^m$ which define how the robot moves to interact with the object after contact. As shown in Fig. 2, to execute an action, the end-effector will first move to a location in

free space near location $a^{loc}$, after which it will interact with the object using the motion parameters $a^m$. After the interaction, the end-effector will move away from the object, a new observation is obtained, and the next action can be taken.

Specifically, given the object point cloud $\mathcal{X}^{obj} = \{x_i \mid i = 1 \ldots N\}$, where $x_i \in \mathbb{R}^3$ are the point locations, the contact location $a^{loc}$ is chosen from among the points in $\mathcal{X}^{obj}$. Thus, $a^{loc}$ is defined over a discrete action space of dimension $N$. We assume a collision-free motion planner to move the gripper to the contact location $a^{loc}$ (see Appendix A.6 for details). In contrast, the motion parameters $a^m$, which de-

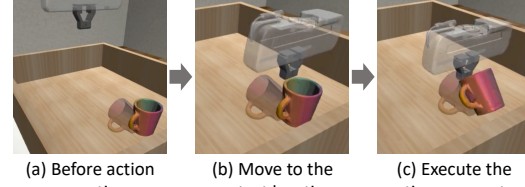

(a) Before action execution    (b) Move to the contact location    (c) Execute the motion parameters

Figure 2: **Illustration of our action space.**

fine how the gripper interacts with the object after contact, are defined in a continuous action space. Furthermore, we define $a^m$ as the end-effector delta position movement from the contact position, hence $a^m \in \mathbb{R}^3$. Our experiments show that translation-only movements are sufficient to enable complex 6D object manipulation in our task. We also include additional experiments on extending motion parameters to enable rotations in Appendix C.4.

The proposed action representation has two benefits compared to previous work. First, it is **temporally-abstracted**. We "abstract" a sequence of lower-level gripper movements of approaching the contact and executing the motion parameters into one action decision step in the RL problem definition. Compared to the common action space of end-effector delta movements [31, 32, 12, 33], the agent with our action space can avoid wasting time learning how to move in free space and instead focus on learning contact-rich interactions. Second, it is **spatially-grounded** since the agent selects a contact location from the observed object point cloud.

## 5.2 Hybrid RL Algorithm

The proposed action space is a hybrid discrete-continuous action space: the contact location $a^{loc}$ is discrete while the motion parameters $a^m$ are continuous. We propose a way to adapt existing off-policy algorithms designed for continuous action spaces [6, 27] to this hybrid action space. First, consider the simpler case of an action space that only has the continuous motion parameters $a^m$. In this case, we can directly apply existing off-policy algorithms as described in Section 3. Given the observation $s$ (which is the point cloud $\mathcal{X}$ in our task), we can train an actor to output the motion parameters $\pi_\theta(s) = a^m$. Similarly, the critic $Q_\phi(s, a^m)$ outputs the Q-value given the observation $s$ and the motion parameters $a^m$; the Q-value can be used to update the actor according to Eqn. 3.

To additionally predict the contact location $a^{loc}$, we also need the policy to select a point among the discrete set of points in the object point cloud $\mathcal{X}^{obj}$. Our insight is that we can embed such a discrete component of the action space into the critic by training the critic to output a per-point Q-value $Q_i$ for each point $x_i$ over the entire point cloud. The Q-value at each point on the object represents the estimated return after selecting this point as the contact location. These Q-values can thus be used not only to update the actor, but also to select the contact location as the point with the highest Q-value. Additionally, we train the actor to also output per-point motion parameters $a_i^m$ for each point $x_i$. If point $x_i$ is selected as the contact location $a^{loc}$, then the motion parameters at this point $a_i^m$ will be used as the gripper motion after contact.

The overall architecture is shown in Fig. 3. The actor $\pi_\theta$ receives as input the point cloud observation and outputs per-point motion parameters $\pi_\theta(\mathcal{X}) = \{a_i^m = \pi_{\theta,i}(\mathcal{X}) \mid i = 1 \ldots N\}$. We call this per-point output an **"Actor Map"**. The critic also receives as input the point cloud observation. It first calculates per-point features $f(\mathcal{X}) = \{f_i \mid i = 1 \ldots N\}$. The critic then concatenates each per-point feature $f_i$ (Section 4) with the corresponding per-point motion parameter $a_i^m$ and inputs the concatenated vector to an MLP. The output of the MLP is a per-point Q-value: $Q_i = Q_\phi(f_i, a_i^m)$, which scores the action of moving the gripper to location $x_i$ and executing motion parameters $a_i^m$. We call this per-point output a **"Critic Map"**. In this way, the critic is able to reason jointly about the contact location (via the feature $f_i$) as well as the motion parameters $a_i^m$. In our implementation, both the actor and the critic use segmentation-style PointNet++ architecture [34] (Appendix B).

At inference time, we can select the point $x_i$ within the object points $\mathcal{X}^{obj}$ with the highest Q-value $Q_i$ as the contact location and use the corresponding motion parameters $a_i^m$. For exploration during training, we define a policy $\pi^{loc}$ which selects the contact location based on a softmax of the

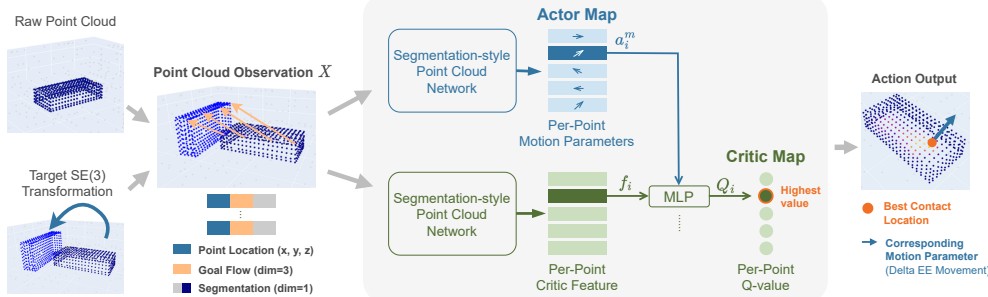

**Figure 3: An overview of the proposed method.** The point cloud observation includes the location of the points and point features. The goal is represented as per-point flow of the object points. The actor takes the observation as input and outputs an **Actor Map** of per-point motion parameters. The Actor Map is concatenated with the per-point critic features to generate the **Critic Map** of per-point Q-values. Finally, we choose the best contact location according to the highest value in the Critic Map and find the corresponding motion parameters in the Actor Map.

Q-values over all of the object points. The probability of a point $x_i$ being selected as the contact location is thus given as:

$$\pi^{loc}(x_i \mid s) = \pi^{loc}(x_i \mid \mathcal{X}) = \frac{\exp(Q_i/\beta)}{\sum_{k=1,\dots,N} \exp(Q_k/\beta)}. \tag{4}$$

$\beta$ is the temperature of the softmax which controls the exploration of the contact location. Note that the background points $\mathcal{X}^b$ are included in the observation $s = \mathcal{X} = \{\mathcal{X}^b, \mathcal{X}^{obj}\}$, but are excluded when choosing the contact location. We modify the update rules of the off-policy algorithm for this hybrid policy. Given $s = \mathcal{X}$, we first define the per-point loss for updating the actor $\pi_{\theta,i}(s)$ at location $x_i$ according to Eqn. 3:

$$J_i(\theta) = -Q_\phi(f_i, a_i^m) = -Q_\phi(f_i, \pi_{\theta,i}(s)). \tag{5}$$

$f_i$ is the feature corresponding to point $x_i$. The total objective of the actor is then computed as an expectation over contact locations:

$$J(\theta) = \mathbb{E}_{x_i \sim \pi^{loc}}[J_i(\theta)] = \sum_i \pi^{loc}(x_i \mid s) \cdot J_i(\theta). \tag{6}$$

$\pi^{loc}(x_i \mid s)$ is the probability of sampling contact location $x_i$, defined in Eq. 4. The difference between Eqn. 6 and the regular actor loss in Eqn. 3 is that we use the probability of the discrete action to weight the loss for the continuous action. To take into account $\pi_{loc}$ during the critic update, the Q-target $y_t$ from Eqn. 2 is modified to be:

$$y = r_t + \gamma \mathbb{E}_{x_i \sim \pi^{loc}}[Q_\phi(f_i(s_{t+1}), \pi_{\theta,i}(s_{t+1}))]. \tag{7}$$

### 5.3 Representing the Goal as Per-Point Goal Flow

As described in Section 4, the objective of our task is to move an object to a given goal pose. Instead of concatenating the goal point cloud to the observed point cloud [17, 18], we represent the goal as per-point "goal flow": Suppose that point $x_i$ in the initial point cloud corresponds to point $x_i'$ in the goal point cloud; then the goal flow is given by $\Delta x_i = x_i' - x_i$. The goal flow $\Delta x_i$ is a 3D vector which is included as the feature of the point cloud observation (concatenated with a segmentation label, resulting in a 4-dimensional feature vector). This flow representation of goal is also used to calculate the reward and success rate for the pose alignment task (Appendix A.3). In the real robot experiments, we estimate the goal flow using point cloud registration (Appendix D). Our ablation experiments suggest that utilizing the flow representation of the goal drastically enhances performance compared to directly concatenating the goal point cloud (Appendix C.1).

## 6 Experiment Setup

We evaluate our method on the 6D object pose alignment task as described in Section 4. The objective is to perform a sequence of non-prehensile actions (i.e. pushing, flipping) to move the object to a given goal pose. In this section, we describe the task setup in simulation, used for training and simulation evaluation (see Section 8 for real robot experiments).

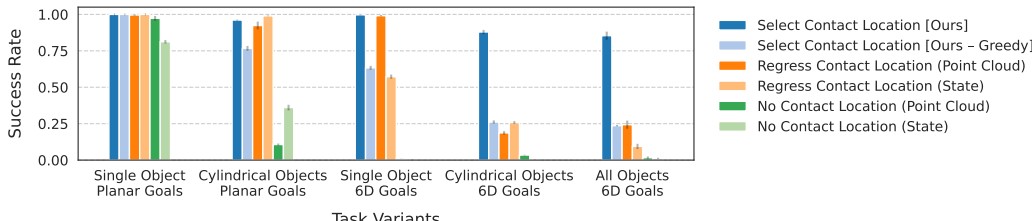

Figure 4: **Baselines and Ablations.** Our approach outperforms the baselines and the ablations, with a larger margin for more challenging tasks on the right. Success rates for simple tasks - pushing a single object to an in-plane goal - are high for all methods, but only HACMan achieves high success rates for 6D alignment of diverse objects.

**Task Setup.** The simulation environment is built on top of Robosuite [35] with MuJoCo [36]. We include 44 objects with diverse geometries from Liu et al. [37]. Details and visualizations of the object models are included in Appendix A.1. The object dataset is split into three mutually exclusive sets: training set (32 objects), unseen instances (7 objects) and unseen categories (5 objects). The 7 unseen instances consist of objects from categories included in the training set, whereas the 5 objects in "unseen categories" consist of novel object categories. An episode is considered a success if the average distance between the corresponding points of the object and the goal is less than 3 cm. More details on our simulation environment setup can be found in Appendix A.

**Task Variants.** To analyze the limitations of different methods, we design the object pose alignment task with varying levels of difficulty. We consider three types of object datasets: An **All Objects** dataset that includes the full object dataset, a **Cylindrical Objects** dataset consisting of only cylindrical objects, and a **Single Object** dataset consists of just a single cube. We also try different task configurations: In the **Planar goals** experiments, the object starts from a *fixed* initial pose at the center of the bin, and the goal pose is a randomized *planar translation* of the starting pose. In the **6D goals** experiments, both the object initial pose and goal pose are *randomized SE(3)* stable poses, not limited to planar transformations. This task requires SE(3) object movement to achieve the goal which imposes challenges in spatial reasoning. These different task variations are used to show at what level of difficulty each of the baseline methods stop being able to complete the task.

# 7 Simulation Results

In this section, we demonstrate the effectiveness of HACMan compared to the baselines and ablations. Fig. 4 summarizes the performance of each method after being trained with the same number of environment interactions. The training curves, tables, and additional results can be found in Appendix C. Implementation details of all methods are included in Appendix B.

**Effect of action representations.** We compare our method with two alternative action representations, summarized in Table 1. In **Regress Contact Location**, the policy directly regresses to a contact location and motion parameters, instead of choosing a contact point from the point

Table 1: Features of the proposed action representation compared to the baselines.

|  | Spatially Grounded? | Temporally Abstracted? |
|---|---|---|
| Select Contact Location [Ours] | ✓ | ✓ |
| Regress Contact Location | ✗ | ✓ |
| No Contact Location [17, 18, 12, 13, 32, 31] | ✗ | ✗ |

cloud as in HACMan. The **No Contact Location** baseline directly regresses to a delta end-effector movement at each timestep. For every action, the robot continues from its position from the previous action, instead of first moving the gripper to a selected contact location. This is the most common action space in manipulation [17, 18, 12, 13, 32, 31]. As input for these two baselines, we use either point cloud observations or ground-truth state, establishing four baselines in total. The baseline that regresses motion parameters from point cloud observations is a common action representation used in prior work in RL from point clouds such as Qin et al. [13]. The baseline that regresses motion parameters from ground-truth state is the most common approach in prior work, such as in Zhou and Held [12] as well as the teacher policies in Chen et al. [17, 18] (see Appendix F for more discussion).

As shown in Fig. 4, these baseline action representations struggle with the more complex task variants. For the most challenging task variant "All Objects + 6D goals," our method achieves a success rate 61% better than the best baseline (see Table 5 for numbers). As mentioned in Section 5, the proposed action representation benefits from being *spatially-grounded* and *temporally-abstracted*. The comparison against the baselines demonstrates the importance of each of these two features

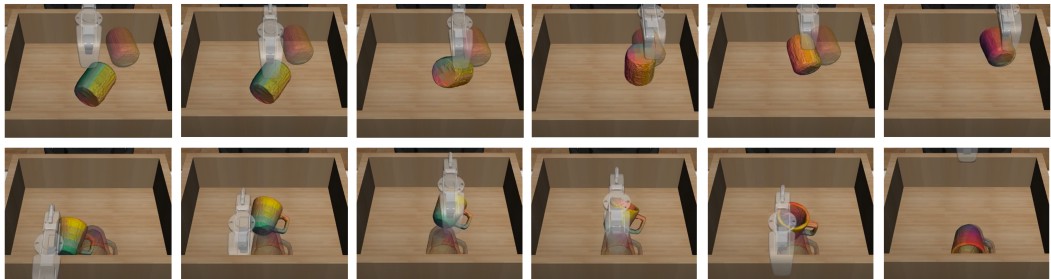

Figure 5: **Qualitative results for the object pose alignment task.** HACMan shows complex non-prehensile behaviors that move the object to the goal pose (shown as the transparent object).

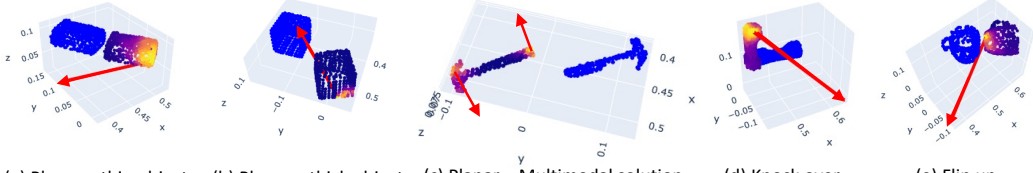

(a) Planar – thin object    (b) Planar – thick object    (c) Planar – Multimodal solution    (d) Knock over    (e) Flip up

Figure 6: **Goal-conditioned Critic Maps.** Blue: goal point cloud. Color map: observed object point cloud. Lighter colors indicate higher Critic Map scores. Red arrows: motion parameters at a selected location. The policy uses different contact locations based on object geometries and goals.

(Table 1). The "Regress Contact Location" baseline still benefits from being temporally-abstracted because the gripper starts from a location chosen by the policy at each timestep; however, this action representation is not spatially-grounded because it regresses to a contact location which might not be on the object surface, unlike our approach which selects the contact location among the points in the point cloud observation. Thus, the "Regress Contact Location" baseline suffers from training difficulties with more diverse objects (last two variants in Fig. 4). The "No Contact Location" baseline is neither spatially-grounded nor temporally-abstracted; it follows the usual approach from prior work [17, 18, 12, 13, 32, 31] of regressing an end-effector delta motion at each timestep. While this is the most common action space in prior work, it has close to zero performance with 6D goals.

**Effect of Multi-step Reasoning.** To test the necessity of multi-step reasoning for the pose alignment task, we experiment with a **"Greedy"** version of HACMan by setting the discount factor $\gamma$ in the RL algorithm to $\gamma = 0$. This forces the algorithm to optimize for greedy actions for each step. Using RL for multi-step reasoning is one of the important differences between our method and previous work such as Where2Act [3] and UMPNet [4] which optimize for one-step contact locations. Fig. 4 indicates that greedy actions might work for planar goals, but suffer from poor performance for 6D goals that requires multi-step non-greedy interactions. For example, the last row in Fig. 5 shows an example of our method pushing the object away from the goal position to prepare for flipping it to the correct orientation, demonstrating non-greedy behavior. In contrast, we find that the greedy ablation often results in local optima of only trying to match the object position but not its orientation.

**Generalization to unseen objects.** The evaluation of our method over unseen objects with 6D goals is summarized in Table 2. Our method generalizes well to unseen object instances and unseen categories without a performance drop. When we increase the maximum episode length

Table 2: **Generalization to unseen objects.**

| Object Set Split | Success Rate | # of Objects |
|---|---|---|
| Train | 0.833 ± .018 | 32 |
| Train (Common Categories) | 0.887 ± .024 | 13 |
| Unseen Instance (Common Categories) | 0.891 ± .033 | 7 |
| Unseen Category | 0.827 ± .047 | 5 |

from 10 steps to 30 steps, our method achieves 95.1% success on unseen categories (Appendix C.7). Table 2 shows that, comparing the same set of object categories ("common categories"), the success rates of the training instances are similar to the unseen instances. The differences in geometry comparing the training objects and the unseen objects are visualized in Appendix A.1.

**Goal-conditioned Object Affordance and Multimodality.** We visualize the Critic Map, which computes the score of each contact location of the object (Fig. 6). The Critic Maps capture goal-conditioned object affordances which describe how the object can be moved to achieve the goal. Fig. 6(a) and (b) are two scenarios of performing translation object motions for different object heights: For a thin object, the Critic Map highlights the region of the top of the object (dragging) or

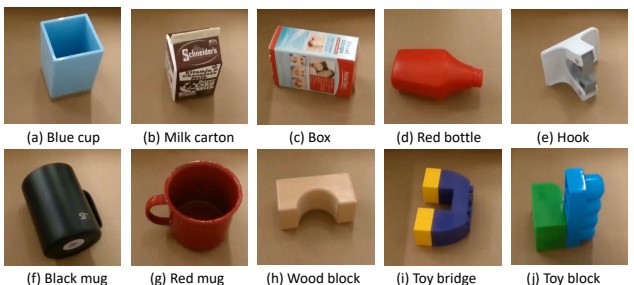

| Object Name | Planar Goals | Non-planar Goals | Total |
|---|---|---|---|
| (a) Blue cup | 4/7 | 4/13 | 8/20 |
| (b) Milk bottle | 6/7 | 10/13 | 16/20 |
| (c) Box | 2/5 | 10/15 | 12/20 |
| (d) Red bottle | 4/7 | 0/13 | 4/20 |
| (e) Hook | 5/8 | 5/12 | 10/20 |
| (f) Black mug | 4/7 | 0/13 | 4/20 |
| (g) Red mug | 5/7 | 3/13 | 8/20 |
| (h) Wood block | 6/7 | 6/13 | 12/20 |
| (i) Toy bridge | 9/10 | 5/10 | 14/20 |
| (j) Toy block | 2/2 | 10/18 | 12/20 |
| **Total** | 47/67 | 53/133 | 100/200 |
| **Percentage** | 70% | 40% | 50% |

Figure 7: **Real robot experiments.** HACMan achieves a $50\%$ overall success rate over unseen objects with different geometries and physical properties, with 6D goal poses.

from the back side of the object (pushing). For a thick object, it prefers to push from the bottom to avoid the object falling over. In Fig. 6(c), the hammer needs to be rotated by 180 degrees. The Critic Map predicts a multimodal solution of pushing from either end of the object. Fig. 6(d) and (e) show out-of-plane motions of the object of knocking over and flipping up the objects.

**Additional Results.** We include additional experiment results in Appendix C, including additional ablations, extending the motion parameters, cluttered scenes, longer training steps, longer episode lengths, and success rate breakdown for each object category.

## 8  Real robot experiments

In the real robot experiments, we aim to evaluate the ability of the trained policy to generalize to novel objects and execute dynamic motions in the real world. We evaluate the policy with a diverse set of objects with different shapes, surface frictions, and densities (Fig. 7). We use random initial poses and random 6D goals (referred to in the previous section as "All Objects + 6D Goals"). For example, the red mug (Fig. 7(g)) has goal poses of being upright on the table or lying on the side. An episode is considered a success if the average distance of corresponding points between the object and the goal is smaller than 3 cm; we also mark an episode as a failure if there is a failure in the point cloud registration between the observation and the goal. Implementation details of the real robot experiments can be found in Appendix D.

Fig. 7 (right) summarizes the quantitative results of the real robot experiments. We run the evaluations without manual reset, which may create uneven numbers of planar versus out-of-plane goals. It achieves $70\%$ success rate on planar goals and $40\%$ success rate on non-planar goals. Non-planar goals are more difficult than the planar goals because they require dynamic motions to interact with the object. A small error in the action may result in large changes in the object movement. Videos of the real robot experiments can be found on the website: `https://hacman-2023.github.io`. The real robot experiments demonstrate that the policy is able to generalize to novel objects in the real world, despite the sim2real gap of the simulator physics and inaccuracies of point cloud registration for estimating the goal transformation. More discussion can be found in Appendix D.

## 9  Limitations

Since the contact location in our action space is defined over the point cloud observation, our method requires relatively accurate depth readings and camera calibration. Further, the contact location is currently limited to the observed part of the object. In addition, for this goal-conditioned task, we represent the goal as per-point flow (Section 5.3) which relies on point cloud registration algorithms. Inaccuracies in the registration algorithm sometimes lead to failure cases in real robot experiments. More discussion of the failure cases can be found in Appendix D.3. In addition, finetuning the policy on the real robot can potentially improve the success rate further.

## 10  Conclusion

In this work, we propose to learn Hybrid Actor-Critic Maps with reinforcement learning for non-prehensile manipulation. The learned policy shows complex object interactions and strong generalization across unseen object categories. Our method achieves a significantly higher success rate than alternative action representations, with a larger performance gap for more difficult task variants. In addition, it would be interesting to explore alternative 3D representations other than point clouds such as implicit representation [38, 39]. We hope the proposed method and the experimental results can pave the way for future work on more skillful robot manipulation over diverse objects.

**Acknowledgments**

We thank Thomas Weng, Ben Eisner, Daniel Seita, Mrinal Kalakrishnan, Homanga Bharadhwaj, Patrick Lancaster, Vikash Kumar, Jay Vakil, and Priyam Parashar for the insightful feedback throughout this project.

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

# Appendix

## Table of Contents

## A Simulation Environment

### A.1 Object dataset preprocessing

We use the object models from Liu et al. [37]. Before importing the object models to Mu-Joco, we perform convex decomposition using V-HACD (https://github.com/kmammou/v-hacd) and generate watertight meshes using Manifold (https://github.com/hjwdzh/Manifold). The objects are first scaled to 10 cm according to the maximum lengths along x, y, and z axis. The object sizes are randomized with an additional scale within $[0.8, 1.2]$ for the "All Objects" task variants.

We filter out a part of the objects in the original dataset due to simulation artifacts such as wall penetration and unstable contact behaviors. For example, some of the long and thin objects can be pushed into the walls and bounce back like springs. Some of the objects cannot remain stable on the table. The filtering procedure proceeds as follows: 1) we drop an object with an arbitrary quaternion

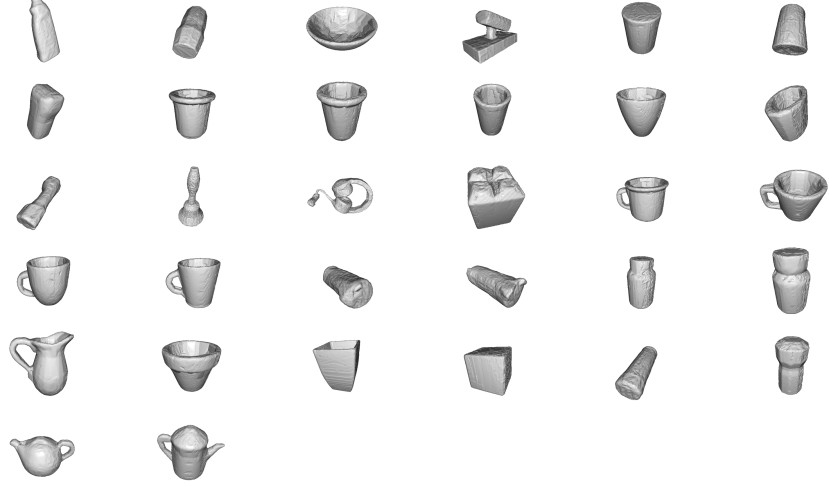

Figure 8: **Training objects.** 32 objects used in training.

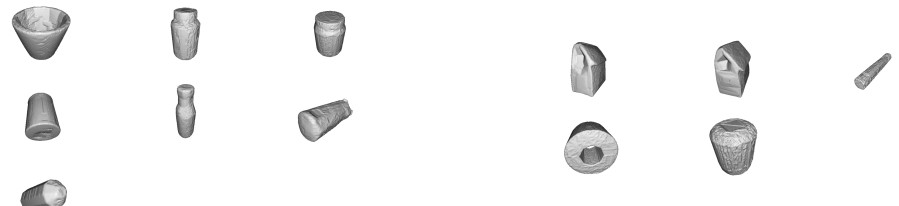

Figure 9: **Evaluation objects (unseen instance).** 7 objects used in unseen instance evaluations. These instances are from the same categories as the training objects.

Figure 10: **Evaluation objects (unseen category).** 5 objects used in unseen category evaluations. They come from 4 randomly chosen categories.

and translation for 100 times; 2) we calculate the percentage of rollouts where the objects are still unstable after 80 simulation steps; 3) we filter out objects with larger than $10\%$ instability rate. We also filter out flat objects because they are hard to flip. Flat objects are defined as objects for which the ratio between the second smallest dimension to the smallest dimension is larger than 1.5. After filtering, we are left with 44 objects. We split the 44 objects into three datasets: train (32 objects), unseen instances (7 objects), and unseen categories (5 objects). The object models of the three datasets are visualized in Fig. 8, Fig. 9 and Fig. 10 respectively. The **Cylindrical Objects** used in the experiments is a subset of the **All Objects** dataset. **Cylindrical Objects** consist of 9 train objects, 3 unseen instance objects, and 4 unseen category objects.

## A.2 Collecting goal poses

To collect stable goal poses, we sample an SE(3) object pose in the air above the center of bin, drop the object in the bin, and then wait until it becomes stable to record the pose. We collect 100 goal poses for each object. At the beginning of each episode, a goal is sampled from the list of stable poses. Furthermore, we randomize the location of the sampled stable goal pose within the bin.

## A.3 Representing the goal as per-point goal flow

As mentioned in Section 5.3, we represent the goal as the "goal flow" of each object point from the current point cloud to the corresponding point in the transformed goal point cloud. In other words, suppose that point $x_i$ in the initial point cloud corresponds to point $x_i'$ in the goal point cloud; then

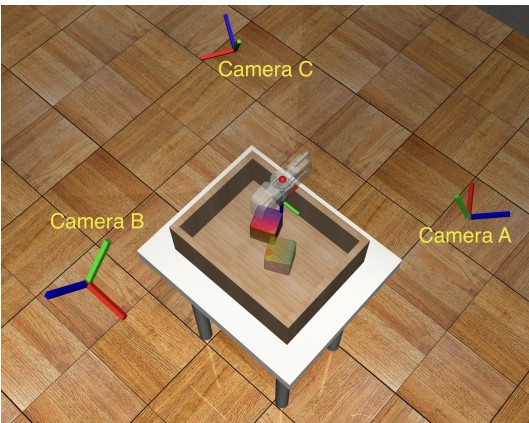

Figure 11: Camera locations in simulation.

the goal flow is given by $\Delta x_i = x_i' - x_i$. The goal flow $\Delta x_i$ is a 3D vector which is concatenated to the other features of the input point cloud to represent the goal. In the ablations in Appendix C.1, we show that such a representation of the goal significantly improves training, compared to other goal representations such as concatenating the goal point cloud with the observed point cloud.

In order to compute the flow to the goal, we need to estimate correspondences between the observation and the goal. In simulation, we calculate the goal flow based on the ground truth correspondences, based on the known object pose and goal pose. In the real robot experiments, we estimate the correspondences using point cloud registration methods (see Appendix D for details).

Further, for training the RL algorithm, we need some measure of the distance between the initial pose and the goal pose as the reward. Rather than computing a weighted average of the translation and rotation distance (which requires a weighting hyperparameter), we instead define the reward at each timestep $r_t$ as the negative of the average goal flow: $r_t = -\frac{1}{N} \sum_{i=1}^{N} ||\Delta x_i||$, in which $|| \cdot ||$ denotes the L2 distance and $\Delta x_i$ is the "goal flow" as defined above. This computation is similar to the "matching score" [40] or "PLoss" [41] used in previous work, except here we use it as a reward function.

### A.4 Success rate definition

An episode is marked as a success when the average distance of the corresponding points between the object and the goal is smaller than 3 cm. More specifically, this is calculated by the average norm of the per-point goal flow vectors as described in Appendix A.3. The episode terminates when it reaches a success. If the episode does not reach a success within 10 steps, it is marked as a failure. We include an additional experiment on longer episode length in Appendix C.7.

### A.5 Observation

The observation space includes a point cloud of the entire scene $\mathcal{X}$. It contains background points $\mathcal{X}^b$ and object points $\mathcal{X}^{obj}$. Note that we move the gripper to a reset pose after every action before taking the next observation. Thus, the gripper is not observed in the point cloud. To get the point cloud, we set three cameras around the bin (Fig. 11). The depth readings from the cameras are converted to a set of point locations in the robot base frame and combined.

The object points are then downsampled with a voxel size of 0.005 m × 0.005 m × 0.005 m and the background points are downsampled with a voxel size of 0.02 m × 0.02 m × 0.02 m. We empirically find that using a slightly denser object point cloud may increase the performance. More specifically, using a 0.005 m × 0.005 m × 0.005 m voxel downsample is slightly better than 0.01 m × 0.01 m × 0.01 m. We suspect that the policy can perform more precise manipulation of the object with a denser point cloud.

After downsampling, we estimate the normals of the object points using Open3D (`http://www.open3d.org/docs/0.7.0/python_api/open3d.geometry.`

`estimate_normals.html`). The estimated normals will be used during action execution (discussed in the next section).

As mentioned in Section 5.3 and Appendix A.3, the feature of each point contains the goal flow and the segmentation mask (foreground vs background). The goal flow of the object point is calculated according to Section A.3. The goal flow of the background point is set to zero. We obtain the segmentation labels of the object points and the background points from Robosuite[35] during simulation. Details of obtaining segmentation labels in real robot experiments are discussed in Appendix D.

### A.6   Action

As mentioned in Section 5, the proposed method uses an action space with a contact location selected from the object points and a set of motion parameters. We discuss the implementation details of executing such an action in the simulation environment in this section. Note that we use a floating gripper as the robot in simulation since we only focus on gripper interactions with the objects.

Once the policy selects a point on the object point cloud, we obtain the corresponding location and estimated normal of the point as described in the previous section. The robot first moves to a "pre-contact" location which is 2 cm away from the contact location along the surface normal. In simulation, this is implemented by directly setting the gripper to the desired pose. In real experiments, we adopt a workaround solution discussed in Appendix D. If the gripper encounters a collision at the desired pose, we mark this action as failure and skip the remaining action execution procedure. After reaching the pre-contact location, the gripper will approach to the desired contact location using a low-level controller.

After that, the robot will execute the motion parameters which is the end-effector delta position command that was output by the policy. For the delta actions, we use an action scale of 2 cm. The delta action is executed with an action repeat of 3. We use Operation Space Controller with relatively low gains to allow compliant contact-rich motions with the object. Note that we only consider translation commands (3 dimensions) without rotation in the main experiments because it leads to sufficiently complex object motion for our task. Appendix C.4 discusses the effect of including rotation in the gripper movements.

The gripper may not exactly reach the desired location in both sim and real, due to the compliant low-level controller and the gripper geometry. We consider this imperfect execution as a part of the environment dynamics. We do not enforce assumptions such as keeping the contact while executing the motion parameter or avoiding other contact points. Avoiding such assumptions on contacts is a strength of the proposed method compared to some of the classical methods [1, 2].

## B   Algorithm and Training Details

### B.1   HACMan (Ours)

HACMan is implemented as a modification on top of TD3 [6] based on the implementation from Stable-Baselines3 (`https://github.com/DLR-RM/stable-baselines3`). We use PointNet++ segmentation-style backbones for both the actor and the critic using the implementation from PyG (`https://pytorch-geometric.readthedocs.io`). Weights are not shared between the actor and the critic. Hyperparameters are included in Table 3. The actor and the critic use the same network size and the same learning rate. To improve the stability of policy training, we clamp the target Q-values according to an estimated upper and lower bound of the return for the task. The location policy temperature $\beta$ is described in Eqn. 4.

### B.2   Baselines

The baselines share the same code framework as HACMan. We discuss their differences with HACMan in this section.

**Regress Contact Location.** Unlike HACMan, this baseline does not use the object surface for contact point selection. Instead, it directly predicts a location (3 dimensions) and a motion parameter

Table 3: Hyperparameters.

| Hyperparameters | Values |
|---|---|
| Initial timesteps | 10000 |
| Batch size | 64 |
| Discount factor ($\gamma$) | 0.99 |
| Critic update freq per env step | 2 |
| Actor update freq per env step | 0.5 |
| Target update freq per env step | 0.5 |
| Learning rate | 0.0001 |
| MLP size | [128, 128, 128] |
| Critic clamping | [-20, 0] |
| Location policy temperature ($\beta$) | 0.1 |

(3 dimensions, represented as a delta end-effector movement). For each action execution, the end-effector moves to the selected location, moves according to the motion parameters, and then resets to the default pose. To improve the performance of this baseline, we project the contact location output to be within the bounding box of the object. Thus, in this baseline, for a location output of the policy, a value of 0 corresponds to the center of the object along a specific dimension, while 1 and $-1$ represent the maximum and minimum boundaries of the bounding box along that dimension, respectively. Since the location output is no longer a point selected from the object surface, we can no longer use the surface normal vector to determine the approach direction as in HACMan. Instead, this baseline always approaches the location from the top at a height equal to the maximum side length of the object bounding box.

**No Contact Location.** This baseline does not use the idea of a contact point. Instead, the policy only predicts a motion parameter (3 dimensions, represented as a delta end-effector movement). For each action execution, the end-effector moves according to the motion parameter starting from where it ends after the previous action, without resetting to the default pose. To reduce the exploration difficulties, we make two additional changes: 1) we always start the end-effector right above the object (at a height equal to the maximum side length of the object bounding box) at the beginning of an episode, and 2) we add an extra term to the reward function that penalizes the end-effector for being too far from the object,

$$J_{\text{dist}} = \begin{cases} -\lambda_{dist}(d_{\min} - 0.05), & d_{min} > 0.05 \text{ m} \\ 0, & otherwise \end{cases} \tag{8}$$

where $d_{min}$ is the minimum distance from the end-effector to the object point cloud vertices, and $\lambda_{dist}$ is the weight for this reward term.

**Point Cloud.** Unlike HACMan, these point cloud baselines use PointNet++ classification-style backbones from PyG (https://pytorch-geometric.readthedoc.io). For each point cloud, it extracts a single global feature vector instead of per-point feature.

**State.** In the state-based baselines, the input consists of the pose of the current object, the goal, and optionally the end-effector if the baseline is using "No Contact Location". Each pose is a vector (dim=7) that consists of a position (dim=3) and a quaternion (dim=4). The model concatenates all the pose vectors into a single vector as the input to an MLP.

We report the best results of the baselines in the paper by searching over different hyperparameters for each baseline, including learning rate, actor update frequency, initial timesteps, and EE distance weight $\lambda_{dist}$. The best hyperparameters for each baseline that are different from HACMan are summarized in Table 4; any hyperparameter not listed in Table 4 is the same as our method (Table 3).

Table 4: Baseline-specific Hyperparameters.

| Baselines | Hyperparameters | Values |
|---|---|---|
| Regress Contact Location (Point Cloud) | Actor update freq per env step | 0.25 |
| No Contact Location (Point Cloud) | Actor update freq per env step | 0.25 |
| | EE Distance Weight $\lambda_{dist}$ | 1 |
| No Contact Location (State) | EE Distance Weight $\lambda_{dist}$ | 5 |

# C  Supplementary Experiment Results

## C.1  Additional ablations

We perform additional ablation studies to analyze each component of the proposed method with all the variants of the object pose alignment task. The results of the ablations are summarized in Fig. 12.

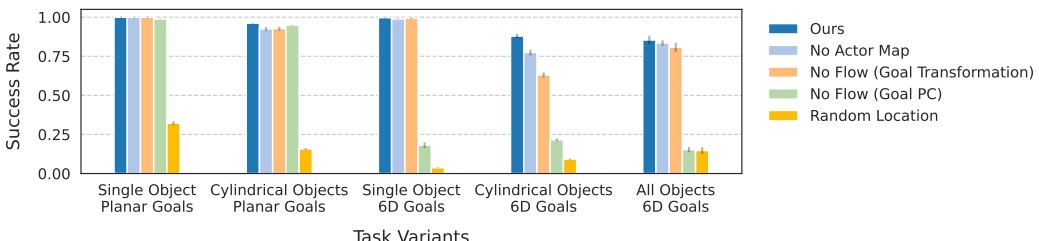

Figure 12: **Additional ablations.** All of the components of our method are essential to achieve the best performance when the task becomes more difficult.

**Effect of Contact Location:** To test the hypothesis that contact location matters for non-prehensile manipulation, we design a **"Random Location"** ablation: the policy randomly selects a contact location on the object instead of learning to predict a contact location. From Fig. 12, we observe a performance drop for not predicting the contact location even for the simplest task variant.

**Effect of Goal Representations:** As described in Section 5.3 and Appendix A.3, in our method, we represent the goal by first computing the correspondence between the observation and goal point clouds and concatenating a per-point "goal flow" to the observation. We include two alternative goal representations to justify the use of goal flow in our pipeline: **"No Flow (Goal PC")** concatenates the goal point cloud with the observed point cloud [17, 18]. We use an additional segmentation label in the point features to distinguish the goal points from the observed points. From Fig. 12, this ablation only works well on planar goals for this task. In **"No Flow (Goal Transformation)"**, we represent the goal as the transformation between the current observation pose and the goal pose. We represent this transformation as a 7D vector that includes a translation vector and a quaternion. We concatenate the 7D goal pose to the observation at all of the object points. Note that, similar to our method, this baseline also requires computing correspondences between the observation and the goal. This approach performs well but slightly worse than our method in the last two task variants.

**Effect of Actor Map:** Instead of using an Actor Map which has per-point outputs, this ablation uses an actor that outputs a single vector of motion parameters while keeping the Critic Map. This is different from the baselines in the previous section that remove both the Actor and Critic Maps. In the **"No Actor Map"** experiments, we observe a relatively minor performance drop compared to the full method. Nonetheless, using the per-point action output from an Actor Map instead of a single output may allow the agent to reason more effectively about different actions for different contact locations, such as the multimodal solution shown in Fig. 6 (middle).

## C.2  Training curves and tables

In this section, we include the full training results for all the methods with additional task variants. Fig. 13 and Fig. 14 include the training curves for the baselines and the ablations. Table 5 and Table 6 are recorded at 200k environment interaction steps from the training curves for all the methods. The numbers in the tables are used to generate the bar plots in Fig. 4 and Fig. 12.

Note we also interpolate between the tasks "Planar Goals" and "6D Goals" and include an additional task configuration with a fixed initial object pose and a randomized 6D goal, "6D Goals (Fixed Init)". This task configuration is combined with the Single Object dataset and the Cylindrical Object dataset. Thus, we include 7 variants in total (5 variants in the main paper).

As discussed in Section 7, the baselines and ablations have poor performance when the task becomes more challenging. Our method achieves the best converged performance across all task variants while being more sample efficient.

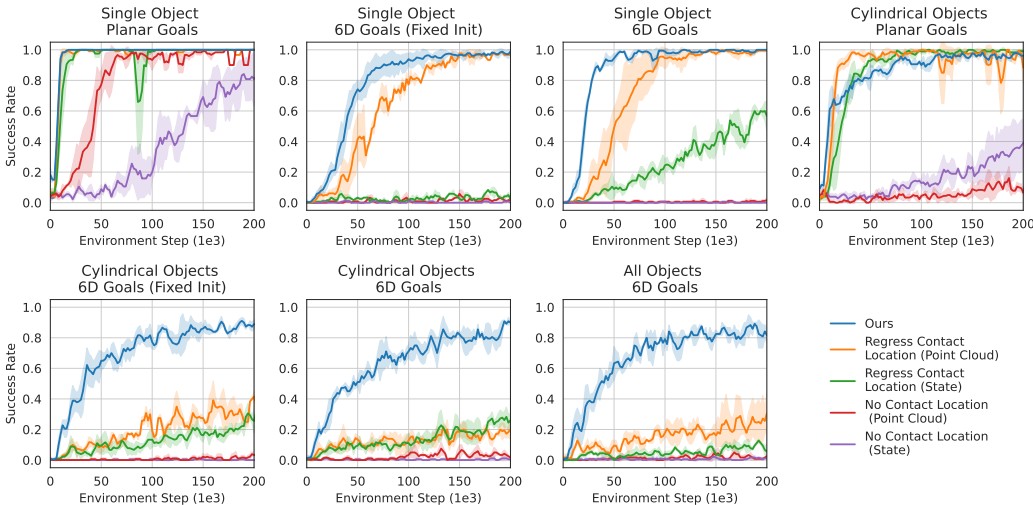

Figure 13: **Baselines.** It shows success rates on the train dataset over environment steps. The shaded area represents the standard deviation across three training seeds.

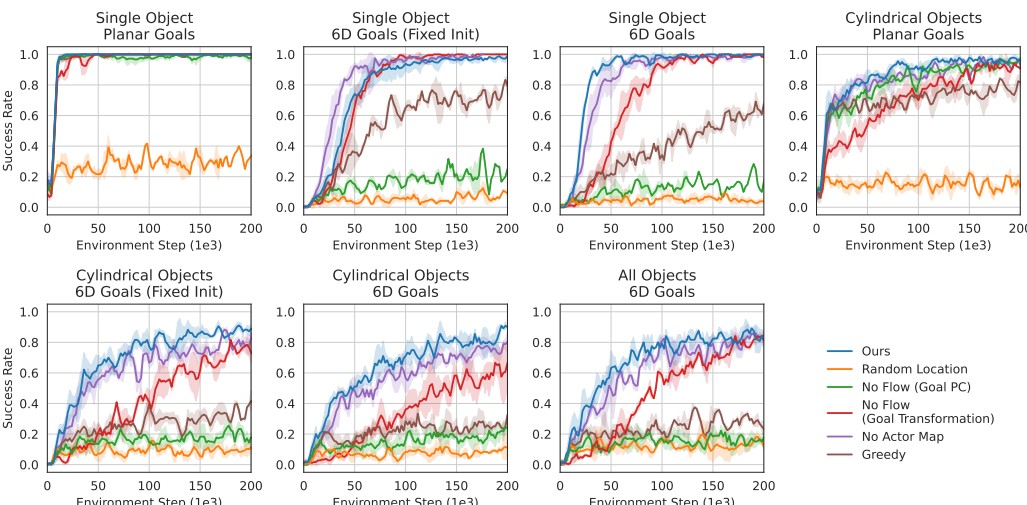

Figure 14: **Ablations.** It shows success rates on the train dataset over environment steps. The shaded area represents the standard deviation across three training seeds.

Table 5: **Baselines.** We compare our method with baselines with different action representations and observations. Our approach outperforms the baselines, with a larger margin for more challenging tasks. The success rate is reported with the mean and standard deviation across three seeds.

| Object Dataset | Task Configuration | No Contact Location | | Regress Contact Location | | Ours |
|---|---|---|---|---|---|---|
| | | State | Point Cloud | State | Point Cloud | |
| Single Object | Planar Goal | **0.812 ± .012** | **0.973 ± .016** | **1.000 ± .000** | 0.996 ± .005 | **1.000 ± .000** |
| | 6D Goal (Fixed Init) | 0.003 ± .000 | 0.020 ± .002 | 0.060 ± .014 | 0.971 ± .005 | **0.982 ± .004** |
| | 6D Goal | 0.000 ± .000 | 0.009 ± .001 | 0.573 ± .015 | 0.991 ± .004 | **0.997 ± .003** |
| Cylindrical Objects | Planar Goal | 0.361 ± .019 | 0.107 ± .007 | **0.990 ± .002** | 0.924 ± .027 | **0.961 ± .003** |
| | 6D Goal (Fixed Init) | 0.001 ± .001 | 0.021 ± .002 | 0.264 ± .017 | 0.324 ± .014 | **0.885 ± .004** |
| | 6D Goal | 0.006 ± .002 | 0.035 ± .002 | 0.258 ± .010 | 0.187 ± .012 | **0.879 ± .014** |
| All Objects | 6D Goal | 0.012 ± .004 | 0.016 ± .009 | 0.094 ± .018 | 0.243 ± .028 | **0.854 ± .028** |

Table 6: **Ablations.** We show that all of the components are essential to achieve the best performance when the task becomes more difficult. Each success rate is reported with the mean and standard deviation across three seeds.

| Object Dataset | Task Configuration | Methods | | | | | |
|---|---|---|---|---|---|---|---|
| | | Random Location | Greedy | No Flow (Goal PC) | No Flow (Goal Pose) | No Action Map | Ours |
| Single Object | Planar Goal | 0.323 ± .011 | **1.000 ± .000** | **0.989 ± .002** | **1.000 ± .000** | **1.000 ± .000** | **1.000 ± .000** |
| | 6D Goal (Fixed Init) | 0.075 ± .005 | 0.754 ± .023 | 0.198 ± .025 | **1.000 ± .000** | **0.991 ± .002** | **0.982 ± .004** |
| | 6D Goal | 0.037 ± .003 | 0.633 ± .014 | 0.181 ± .018 | **0.994 ± .004** | **0.989 ± .002** | **0.997 ± .003** |
| Cylindrical Objects | Planar Goal | 0.158 ± .006 | 0.767 ± .017 | **0.949 ± .003** | 0.927 ± .012 | 0.925 ± .012 | **0.961 ± .003** |
| | 6D Goal (Fixed Init) | 0.097 ± .006 | 0.346 ± .012 | 0.189 ± .009 | 0.746 ± .015 | 0.805 ± .016 | **0.885 ± .004** |
| | 6D Goal | 0.093 ± .004 | 0.262 ± .011 | 0.216 ± .008 | 0.631 ± .016 | 0.775 ± .018 | **0.879 ± .014** |
| All Objects | 6D Goal | 0.147 ± .021 | 0.293 ± .026 | 0.153 ± .017 | 0.808 ± .028 | **0.835 ± .017** | **0.854 ± .028** |

## C.3   Additional baseline: Global feature with query contact location

We consider an additional baseline in this section where both the actor and the critic use a global point cloud feature and a query contact location as input. The query contact location is represented as a 3D coordinate $(x, y, z)$. More specifically, the actor takes as input a global feature and a contact location and outputs a continuous vector of motion parameters. The critic takes as input a global feature and a contact location and outputs a Q-value. Since both the actor and the critic require a contact location as input, we still need a way of selecting the query contact location. We follow a similar way as our method to use the observed points on the object as candidate queries and select the contact location based on the highest Q-value. In this way, the action space remains a discrete-continuous action space as our method, but it uses a global point cloud feature rather than a segmentation-style per-point feature.

As shown in Fig. 15, this alternative baseline performs worse than our method. We hypothesize that a segmentation-style point cloud network can reason about the local point features more effectively than a global feature extractor due to skip connections.

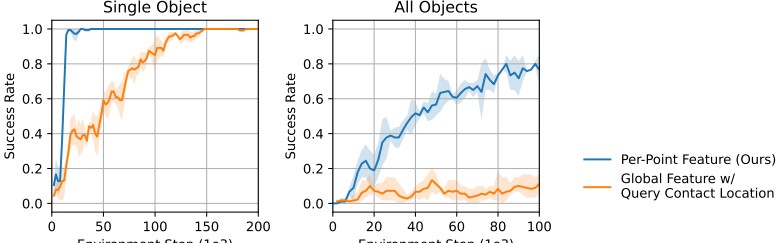

Figure 15: **Comparison between our method and the additional baseline with query contact locations.** The left figure shows the success rate of the simplest task variant - a single object with planar goals. The right figure shows the most challenging task variant - all objects with 6D goals. The shaded area represents the standard deviation across three training seeds. Our method performs better than the baseline in both cases.

## C.4   Extending Motion Parameters

The motion parameters in the main results are defined as a 3D vector that describes the translation motion of the gripper. In this section, we extend the motion parameters in different ways:

**6D Contact.** The motion parameters also predict the orientation of the gripper when the gripper approaches the contact location. The orientation is in the form of ZYX Euler angles. To account for the physical constraints of our task setup, we restrict the y and x angles to the range of $[-0.5\pi, 0.5\pi]$, and the z angle to the range of $[-\pi, \pi]$.

**6D Motion.** We introduce the ability for the gripper to change orientation while executing the motions after making contact. Similar to the translation motion parameters, the rotation motion parameters (ZYX Euler angles) represent the delta rotation at each action repeat step.

**Per-point Contact Location Offset.** We conduct an experiment where the agent learns a per-point contact location offset combined with 3D motion. The agent's continuous action space is defined as

(contact_offset, 3D motion parameters). For each action on a given point at location $x$, the agent uses $(x + x_{\text{offset}})$ as the contact location. Notably, the $x_{\text{offset}}$ value is mapped to scale with the bounding box since it ranges between $[-1, 1]$.

Table 7 compares the performance of HACMan with the modified action spaces. The success rates are reported along with their corresponding standard deviations. We find that including 6D motion in the motion parameters results in a slightly higher success rate. However, 6D contact or contact offset does not seem to provide significant benefits. We also try to include 6D motion in the **Regress Contact Location** baseline. As shown in Table 7, 6D motion improves the performance of the baseline, but the success rate is still much worse than our method.

Table 7: **Success rates with different motion parameters.** All methods are evaluated on all train objects with 6D goals.

| Method | Success Rate |
|---|---|
| HACMan Default | $0.833 \pm .018$ |
| + with 6D Motion | $0.866 \pm .090$ |
| + with 6D Contact | $0.819 \pm .077$ |
| + with Contact Offset | $0.800 \pm .011$ |
| Regress Contact Location Default | $0.243 \pm .028$ |
| + with 6D Motion | $0.356 \pm .133$ |

## C.5 Experiments in cluttered environments

We can directly apply HACMan to a setting of manipulating objects in cluttered scenes. We conduct preliminary experiments in which we introduce varying numbers of scene objects into the bin. The scene objects serve as obstacles that add challenges to the task. We train HACMan under two conditions: **with one scene object** and **with five scene objects**, and we compare the results with the performance achieved in the absence of any scene objects

Table 8: **Success rates under different cluttered scenes.** All methods are evaluated with 6D goals.

| # of Scene Objects | Success Rate |
|---|---|
| 0 (Default) | 0.833 |
| 1 | 0.773 |
| 5 | 0.580 |

(default setting). From Table 8, as expected, the task becomes more challenging when there are more obstacles in the bin. As illustrated in Fig 16, the policy tends to push the object directly toward the goal by pushing the scene object aside.

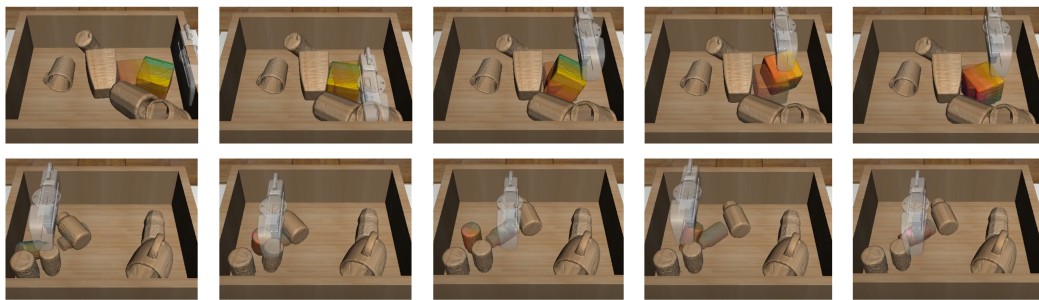

Figure 16: **Qualitative results for object pose alignment tasks in cluttered environments.** HACMan shows complex non-prehensile behaviors that move objects to goal poses (shown as the transparent objects). The scene objects are colored in brown to distinguish from the target object to be manipulated to the goal pose.

## C.6 Effect of longer training time

Although we report the success rate at 200k training steps for all the results due to computational limitations, our method continues to improve performance with longer training (Figure 17). The graph illustrates the success rate achieved by our method as the number of training steps increases. Notably, after 500k training steps, our method achieves a success rate of $91.1 \pm 7.3\%$, significantly improving from the $83.3\%$ success rate reported in the main text (at 200k training steps).

## C.7 Effect of longer episode lengths

We conducted an additional experiment to explore the relationship between success rates and maximum episode length. In the main paper, our episodes were limited to a maximum of 10 steps, and

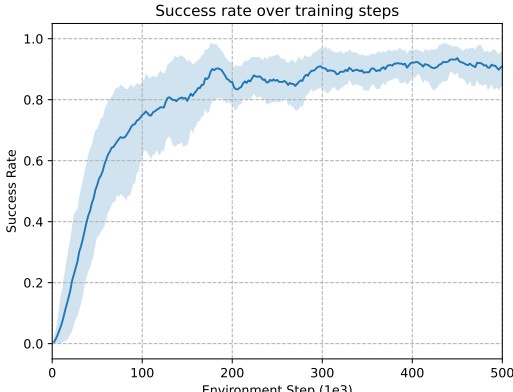

Figure 17: **Success rate with extended training.** The success rate of our method reaches $91.1 \pm 7.3\%$ after 500k training steps, compared to 83.3% after 200k training steps.

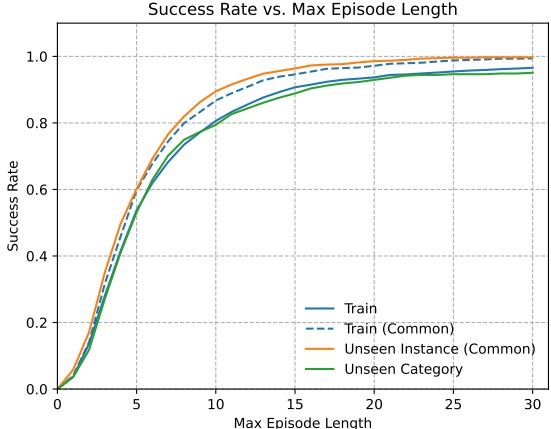

Figure 18: **Success rates at various maximum episode lengths.** This line plot shows the success rates of HACMan evaluated on the four datasets. It is worth noting that the success rates for Unseen Instance (Common) and Train (Common) are marginally higher compared to Train and Unseen Category, similar to the pattern in Table 2.

any episode exceeding this limit was deemed a failure. During this additional evaluation, we relaxed the episode length restrictions and allowed the agent to operate with a maximum episode length of 30. As shown in Fig 18, HACMan achieves more than $95\%$ success rates across all datasets (Train 96.6%, Train (Common) 99.4%, Unseen Instance (Common) 99.7%, Unseen Category 95.1%) when the maximum episode length is extended to 30. This suggests that providing the agent with a longer time horizon enables it to achieve higher success rates without the need for retraining.

### C.8 Per-category result breakdown

Fig. 19 shows the breakdown of the results for each object category. Although our method demonstrates consistent performance across the majority of objects, there are certain objects with geometries that pose intrinsic challenges for our approach. For example, our method is limited to poking a bowl from the top due to occlusions, making it difficult to flip an upward-facing bowl downwards.

### C.9 Final Distance to Goal

To further analyze the performance of our method, Fig. 20 visualizes the distribution of distances to goal of our method at the end of the episodes for the "All Objects 6D Goals" task variant. These distances are computed as the norms of the flow distances between the objects and their respective goals.

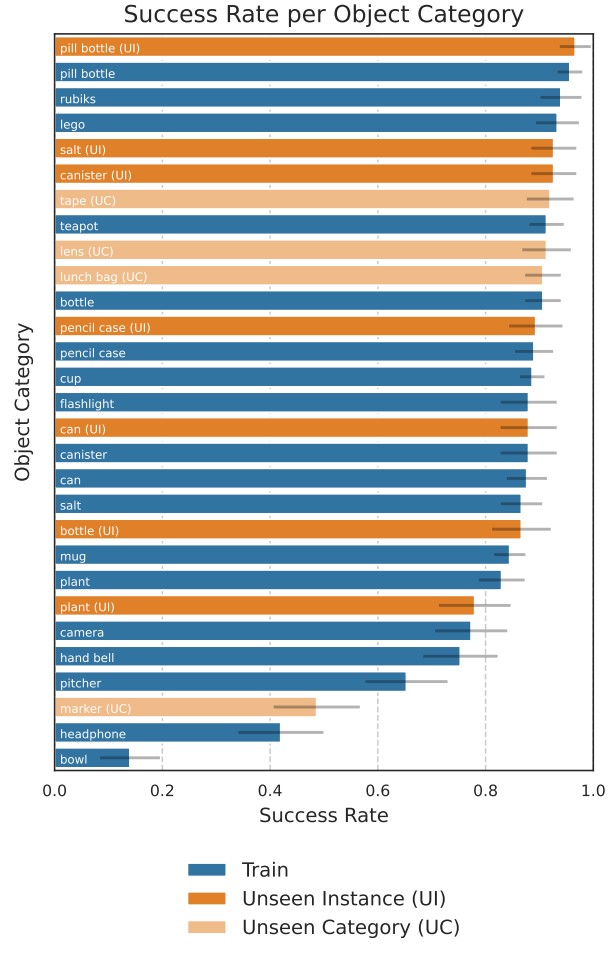

Figure 19: **Results breakdown.** Object categories in the unseen instance set (orange) can be compared to the same object categories in the train set (blue) to see the level of instance generalization.

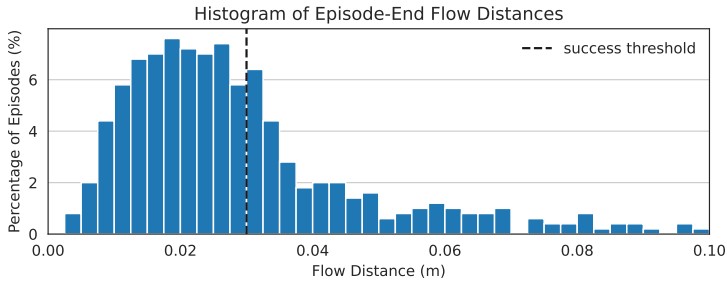

Figure 20: **Distribution of distances to the goal at the end of the episode for our method in the "All Objects 6D Goals" experiment.** The vertical dashed line represents the success threshold at a distance of 0.03m. The distribution has a median of 2.57cm, a mean of 3.66cm, and a standard deviation of 4.27cm.

## D  Real Robot Experiments

### D.1  Real robot setup

The robot setup is shown in Fig. 21. We use three cameras on the real robot to get a combined point cloud. We follow a similar procedure as Appendix A to process the point cloud and to execute the action except the following details: We segment the object points from the full point cloud

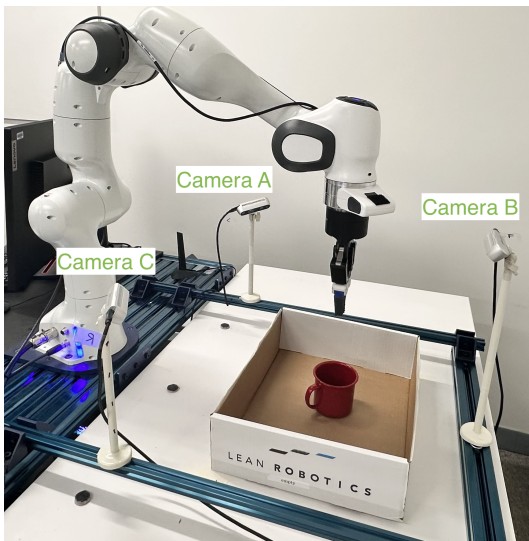

Figure 21: Real robot setup.

based on the location and the dimension of the bin instead of using the ground truth segmentation labels from Robosuite. To move the gripper to the pre-contact location, we first move the robot to a location above the pre-contact location and then move down to the pre-contact location, instead of "teleporting" the gripper in simulation.

To obtain goals for the real world evaluation, we record 10 goal point clouds for each object by manually setting the objects into different stable poses. During each timestep, we use point cloud registration algorithm to estimate the goal transformation to calculate the goal flow. Specifically, we use the global registration implementation from Open3D (http://www.open3d.org/docs/release/tutorial/pipelines/global_registration.html) and then use Iterative Closest Point (ICP) for local refinement. Note that we only match the shapes of the object instead of matching both the colors and the shapes due to the limitation of the registration algorithms.

Note that the evaluation process can be done automatically without any manual resets. The reward and the episode termination condition (Appendix A.4) are both calculated automatically.

For the real robot experiments, we use the policy trained with "6D goals" and the "All Objects" dataset. We perform zero-shot sim2real transfer without finetuning or additional domain randomization. We have tried to add noise to the contact location execution and add noise to the point cloud observation. However, these modifications did not result in better real robot performance.

## D.2 Analysis

We include additional analysis on the real robot results in this section. The proposed method assumes an estimated goal transformation as input. To estimate the transformation from the object to the goal, we use point cloud registration, as described above. However, the estimation of the transformation might not be perfect in the real world. To better understand the performance of our system, we define two types of evaluation criteria: The "flow success" is automatically calculated based on the *estimated* point cloud registration according to the evaluation metric in Appendix A.4. Hence, "flow success" will sometimes mark an episode as a success or failure incorrectly due to errors in the point cloud registration. For the "actual success" evaluation metric, we manually mark as failures the cases among the flow success episodes where the goal estimation is significantly wrong. Thus, "flow success" indicates the performance of the trained policy (assuming perfect point cloud registration at termination) while the "actual success" indicates the performance of the full system (accounting for errors in the point cloud registration). Fig. 7 in the main text reports the actual success. We include both success metrics in Table 9 below. The policy achieves a 61% success rate based on the flow success, indicating that some of our errors are due to failures in point cloud registration.

Table 9: **Additional analysis on the real robot experiments.** An episode is considered a "flow success" if the average norm of the estimated flow is less than 3 cm. An episode is considered as an "actual success" if the object is aligned with the goal pose without point cloud registration failure.

| Object Name | Planar Goals | | Non-planar Goals | | Total | |
|---|---|---|---|---|---|---|
| | Flow | Actual | Flow | Actual | Flow | Actual |
| (a) Blue cup | 4/7 | 4/7 | 7/13 | 4/13 | 5/20 | 4/20 |
| (b) Milk carton | 6/7 | 6/7 | 10/13 | 10/13 | 16/20 | 16/20 |
| (c) Box | 2/5 | 2/5 | 10/15 | 10/15 | 12/20 | 12/20 |
| (d) Red bottle | 7/7 | 4/7 | 6/13 | 0/13 | 13/20 | 4/20 |
| (e) Hook | 5/8 | 5/8 | 5/12 | 5/12 | 10/20 | 10/20 |
| (f) Black mug | 4/7 | 4/7 | 2/13 | 0/13 | 6/20 | 4/10 |
| (g) Red mug | 5/7 | 5/7 | 7/13 | 3/13 | 12/20 | 8/20 |
| (h) Wood block | 6/7 | 6/7 | 8/13 | 6/13 | 14/20 | 12/20 |
| (i) Toy bridge | 9/10 | 9/10 | 7/10 | 5/10 | 16/20 | 14/20 |
| (j) Toy block | 2/2 | 2/2 | 10/18 | 10/18 | 12/20 | 12/20 |
| **Total** | 50/67 | 47/67 | 72/133 | 53/133 | 122/200 | 100/200 |
| **Percentage** | 75% | 70% | 54% | 40% | 61% | **50%** |

### D.3 Failure cases

We discuss the failure cases of the real robot experiments in this section and include the videos on our website: https://hacman-2023.github.io/. The most noticeable failure cases are due to the errors of point cloud registration. The challenges of the registration methods come from noisy depth readings and partial point clouds. The error of the point cloud registration methods will lead to unexpected actions during the episode. In addition, it may end the episode early because the episode termination depends on the goal estimation. This motivates us to separate out the success criteria in Table 9 based on the failures of the registration method.

On the action side, both the contact location and the motion parameters might have execution errors. Since the contact location is selected from the observed point cloud, when the camera calibration is not accurate enough, the robot might not be able to reach the desired contact location of the object. In addition, since we use a compliant low-level controller to execute the motion parameters, the robot might not be able to execute the desired motion the same way as in the simulation.

In addition, the object dynamics might be different from the simulation due to the surface friction and the density of the object. The performance of our method could be further improved with domain randomization over the physical parameters.

## E  More discussion on non-prehensile manipulation

Non-prehensile manipulation is an important aspect of the robot's capabilities, particularly in scenarios where grasping encounters limitations, as demonstrated in Fig. 22. This section discusses the importance of non-prehensile actions in two key contexts:

**Environment Occlusion.** The first row of Figure 22 shows an example where the potential grasp poses of the object are occluded by the wall. Non-prehensile moves, like nudging objects to a better position, offer a practical solution in such a scenario.

**Oversized Objects.** The last two rows of Figure 22 include scenarios where certain dimensions of the object are larger than the width of the gripper.

## F  More discussion on the related work

### F.1  Compared to Chen et al. [17, 18]

Our work is substantially different from Chen et al. [17, 18] from the follow aspects:

**Approach:** The approach in Chen et al. [17, 18] follows a student-teacher training pipeline. The teacher training is equivalent to the "No Contact Location" baseline with "states" observations in our paper. The policy takes all the relevant robot state and object state information and outputs delta

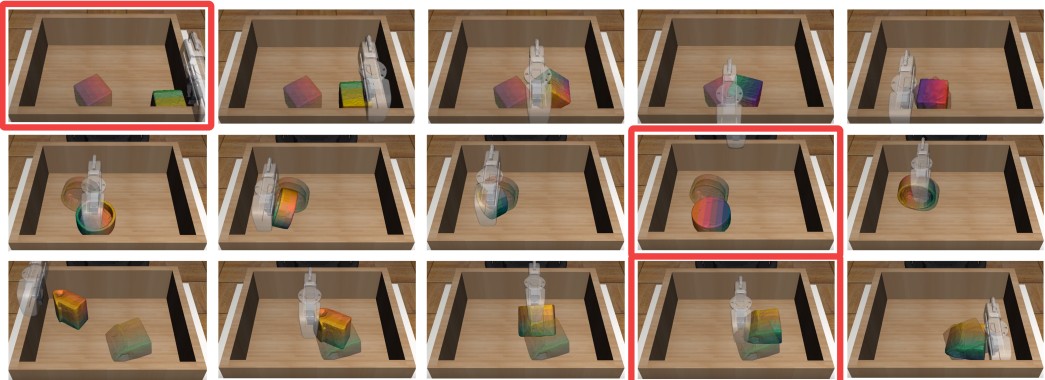

Figure 22: **Examples showcasing limitations of prehensile manipulation.** The frames where prehensile manipulation is challenging are highlighted. The first row shows a cube placed at the corner of the bin, where any grasp is obstructed by the bin wall. Both the second and third rows depict instances where objects are too large to be grasped at specific poses.

robot actions. Note that they train a single teacher policy across all shapes without using the point cloud which results in a state-observation policy that is "robust" to shapes instead of "adaptive" to shapes (see Discussion section in Chen et al. [18]). As shown in Table II, this baseline performs significantly worse than our method in our task because it lacks shape information from the point cloud and the robot-centric action space is not as efficient as our object-centric action space. On the other hand, although the student policy in Chen et al. [17, 18] takes point cloud observation, it is trained using imitation learning from the teacher, so its performance is upper bounded by the teacher policy which has been shown to be worse than our proposed method.

**Task:** We investigate a completely different task and thus the numbers are not really comparable with the numbers from previous work [17, 18]. First, we use a simple gripper instead of a dexterous hand. Second, we consider matching the orientation and position of the goal pose while Chen et al. [17, 18] only considers orientation.

### F.2   Compared to Cheng et al. [1], Hou and Mason [2]

Unlike Cheng et al. [1], Hou and Mason [2], our method is not limited to a simplified gripper model and does not require the knowledge of object environment contact modes which are challenging to estimate during real robot execution. In addition, Cheng et al. [1], Hou and Mason [2] are restricted by quasi-static assumptions. Although our method requires static point cloud observations in between robot actions, the robot interaction with the object is not restricted to quasi-static motion. As shown in Figure 23, the flipping motion could be non-quasi-static. However, we also want to point out that static point cloud observations may limit the method from more complex dynamic motions.

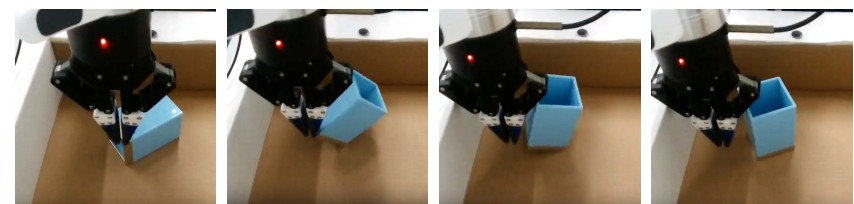

Figure 23: **An example of non-quasi-static motion.** The figure shows an example of executing the motion parameters to flip a mug upright. After the gripper pushes against the edge of the mug (first two images), it relies on the inertia of the mug to finish the motion which is not quasi-static (last two images).

