# OpenReview forum: "HACMan: Learning Hybrid Actor-Critic Maps for 6D Non-Prehensile Manipulation"
_robot-learning.org/CoRL/2023/Conference — CoRL 2023 Oral_

### Official Review · Reviewer_3gnN · 2023-07-19

**Confidence:** 5
**Originality:** Fair
**Technical Quality:** Excellent
**Clarity Of Presentation:** Good
**Impact:** 3

**Recommendation:**

Weak Accept: I recommend accepting the paper, but will not argue for my recommendation if the majority of other reviewers have a different opinion.

**Review:**

Strength:
- This work specifically targets the repositioning of single objects via non-prehensile manipulation. The main contribution lies in exploring how the design of the action space influences data efficiency for such manipulation. The authors propose an action space that is temporally abstracted and spatially grounded. This means that a single action extends over various lengths of time and is associated with specific locations in physical space. Specifically, the authors suggest using an object-centric action representation to simplify the manipulation. This approach constrains the robot's actions to translational movements upon contact with the object. The authors also propose to represent the goal as a "goal flow" within the object's point cloud. This goal flow is determined by the difference between the initial point and the corresponding point in the goal point cloud. In the simulation experiment, the authors illustrated how each of these three components enhances the robot's performance in the task of non-prehensile manipulation of a single object. They also demonstrated how their proposed action and goal representations significantly bolster learning efficiency for 6D object repositioning tasks within a reinforcement learning process. The proposed concepts have been well executed and substantiated with extensive experimental results.

Weakness:
- A primary limitation of this work is its narrow focus on a very specific task space -- table-top single object repositioning. While the authors provide extensive experimental results proving the efficacy of their proposed action and goal representation for this particular task, these results aren't surprising. This is because the representations are specifically designed to simplify the learning problem in this context. Consequently, the paper does not present sufficient evidence to support the claim made in line 338, which suggests that this action space could be extended to a wider variety of manipulation.' The reviewer posits that the assumptions inherent in the proposed action space might even lead to decreased performance in other more complex manipulation tasks. For instance, in a multi-object repositioning scenario, an action not spatially grounded to a single target object could potentially move two or more non-contacting neighboring objects toward their goal poses in a single action, thereby reducing the total control effort.
- The reviewer finds the most intriguing aspect of this paper to be the exploration of how a temporally-abstracted and/or spatially-grounded action space can improve learning efficacy. However, while the action space proposed in the study exemplifies these characteristics, it does not fully represent them. Future research could benefit from comparing different action spaces that incorporate either or both of these traits. For instance, it would be interesting to investigate what might happen if the temporally-abstracted action involved a more complex set of carefully designed parameterized motion primitives.
- Without a comparison to other baseline methods, the real-robot experiment offers limited useful information. Considering the relatively high tolerance for imprecision in the success condition (i.e., an average distance of 3 cm between corresponding points of the object and the goal), a 70% success rate on planar goals and a 40% success rate on non-planar goals is not particularly impressive or informative. The experiment would have been more interesting if it had compared the impact of the proposed action space on zero-shot transfer to at least one baseline method.

**Quality Of The Limitations Section:**

Additional details required

**Questions For Rebuttal:**

- Is the feature f_i mentioned in line 187 just the goal flow discussed in sec 5.3? If so, it would be clearer to at least mention it in Sec.5.2.
- The action space in the "regress contact" baseline constrains the end-effector's orientation to always point straight downward. The loss of certain degrees of freedom regarding the gripper orientation could potentially affect the robot's performance. Would allowing the policy to specify the gripper orientation in such a baseline action space provides a fairer comparison with the proposed action space?

**Robotics Focus:**

Sufficient demonstration on hardware

**Summary Of Paper:**

This paper examines the challenges inherent in non-prehensile robotic manipulation. The authors utilize an object-centric action representation to simplify the process by constraining the robot's actions to translational movements upon contact with the object. The robot selects a contact point and a set of motion parameters, which determine the trajectory post-contact, based on the observed object point cloud. This approach enables the robot to focus on learning the intricacies of the contact-rich segments of the action. This action representation is then integrated into an actor-critic reinforcement learning framework. The contact location is defined in a discrete action space and the motion parameters in a continuous action space. The actor-network generates continuous motion parameters on a per-point basis, while the critic network predicts per-point Q-values over the object point cloud.

**Summary Of Recommendation:**

This work offers substantial experimental details to demonstrate that the proposed problem representation performs effectively for tabletop single-object repositioning tasks. These implementation details will undoubtedly prove beneficial to the community. However, the current problem setup lacks novelty and offers limited insights that can be generalized to more complex tasks, as described in the Review section. Therefore, the reviewer deems this work as a borderline accept.

---

### Official Review · Reviewer_sYWT · 2023-07-19

**Confidence:** 4
**Originality:** Excellent
**Technical Quality:** Excellent
**Clarity Of Presentation:** Excellent
**Impact:** 4

**Recommendation:**

Strong Accept: I recommend accepting the paper and will argue for my recommendation even if other reviewers hold a different opinion.

**Review:**

### Strengths:
1. This paper tackles a challenging and widely useful task: general 6-DoF manipulation without grasping.
2. The chosen action representation is clever and novel, making it possible to learn this task with pure RL autonomously, without requiring demonstrations or multi-stage training as in prior work. This action representation, combined with this RL method, may have significant impact beyond non-prehensile manipulation.
3. Training with a hybrid discrete and continuous action space is non-trivial, but this method resolves this in a neat way. Weighting the actor loss for the continuous action by the probability of selecting the discrete action is quite elegant and intuitive.
4. The experiments are well-designed and answer interesting research questions. For example, a baseline which follows the reward greedily performs less well (Figure 4), showing the importance of the multi-step reasoning capabilities of this RL method. The Critic Map visualisations showing how the method captures multi-modal distributions (often difficult for alternative methods which use behavioral cloning to bootstrap from demonstrations) are great to see. Some of the results are quite surprising as well as impressive (e.g. on generalization to new objects).

### Weaknesses:
1. The paper may benefit from slightly stronger motivation for non-prehensile manipulation, compared to grasping approaches. Intuitively, I am persuaded by the paper’s Introduction which says that non-prehensile interaction is important for human dexterity, e.g. nudging an object into place when grasping is difficult due to clutter. However, the experiments in the paper use a clear workspace: that is, they do not yet show non-prehensile manipulation in the setting where it is most needed, compared with grasping. A small demo in a more cluttered scene where grasping fails but non-prehensile manipulation succeeds may help the paper to persuade more people in the community to build on this approach and work on this kind of manipulation.
2. In the experiments, the paper uses a 3cm error margin to determine whether an episode succeeds. This is quite reasonable in general, but it would also be interesting to see exactly how precise this non-prehensile manipulation can be (e.g. by also reporting the median error, for one experiment). This is because humans usually use this non-prehensile manipulation for millimetre-level “nudging”, and so precision seems quite important for this approach.
3. The approach uses only depth measurements, and no RGB, which may help with sim2real but also means that it may be difficult to get this to work for objects where depth measurements are poor, such as highly reflective or thin or transparent objects (as the authors discuss in the limitations section).
4. Nevertheless, a sim2real gap remains for this method, likely due to differences in contact dynamics between the simulator and real objects. This leads to a drop in success rate (down to 40% success rate in the real world on non-planar goals). This is perhaps the main limitation / bottleneck in the method’s performance in the real world. However, the simulation performance and the complex beheviors exhibited are very impressive.

### Minor issues:
1. The phrase “temporally abstracted” is an intriguing concept but the link between the literal meaning of the words in the phrase and what it means in this context could be made clearer. If I understand correctly, temporally abstracted here means that the action is not tied to a specific timestep in seconds, since the robot can move around in free space for as long as it needs before reaching the right point and executing this contact action. So, an extra explanation along these lines which refers to timesteps could be useful in justifying the name “temporal abstraction”. Otherwise, the “temporal” nature of this concept is not so clear.
2. One point which may benefit from a little more explanation for clarity is this: in the preliminaries, the policy pi_theta is deterministic (Line 109), but nevertheless during exploration a stochastic policy is still used as in Equation 4 (since this is an off-policy method). Just flagging this distinction (that a different, stochastic policy may still be used for exploration) in the Preliminaries may be helpful, when it is mentioned that the policy is deterministic.

### Questions for rebuttal:
1. Can this same method be used for prehensile manipulation? A single predicted contact point is already immediately applicable for suction grippers. Have the authors considered whether future work can extend this method to predicting contact locations for parallel-jaw gripper? This would be helpful to better assess the full impact of this work on future research.
2. It is fortunate but surprising that as per Table 2, there is no performance drop for unseen object instances or even new categories. This strong generalization is particularly interesting because there are not that many objects in the training set (32). Intuitively, one might expect that the new test-time point clouds observed would be quite unlike anything seen by the point cloud networks during training, and so they might give strange predictions. Would the authors be able to shed some more light on what makes this possible? Is there an ablation study which could be conducted to further explore which components contribute more to this strong generalization performance?
3. For future work, have the authors considered using this non-prehensile manipulation as a second stage after pick-and-place, in order to “nudge” an object into place very precisely?
4. It seems like the paper focuses on object rearrangement tasks, with rigid objects. The method could theoretically be extended to articulated objects. What would be required to make this possible?
5. The experimental setup seems suitable for quite sample-efficient, reset-free real world RL. Have the authors considered this approach? Maybe fine-tuning on real world episodes would help further close the sim2real gap.



**Quality Of The Limitations Section:**

Limitations are addressed clearly

**Questions For Rebuttal:**

Please see the main review for questions for rebuttal.

**Robotics Focus:**

Sufficient demonstration on hardware

**Summary Of Paper:**

The paper addresses a task where the robot must manipulate an object into a 6-DoF goal pose without grasping it. To address this, the proposed method incorporates a novel action representation which selects an interaction point and predicts a gripper movement to perform this interaction, and integrates this into an RL framework. The paper’s claims are extensively validated through simulation benchmarking, a range of ablation studies, qualitative visualisations, and real robot experiments.

**Summary Of Recommendation:**

This paper proposes a novel and sensible idea for action parameterization, and integrates it elegantly into an RL algorithm. The results are impressive, particularly for multi-step, difficult, non-planar manipulation sequences. The experiments also contain some valuable and surprising insights, e.g. on generalization to novel objects. The paper could be further improved through additional techniques which reduce the sim2real gap for this task, in order to improve real-world performance. Overall, this paper would be a great contribution to CoRL and will be widely appreciated by the community for its insights.

---

### Official Review · Reviewer_h5wU · 2023-07-20

**Confidence:** 5
**Originality:** Very Good
**Technical Quality:** Very Good
**Clarity Of Presentation:** Excellent
**Impact:** 4

**Recommendation:**

Strong Accept: I recommend accepting the paper and will argue for my recommendation even if other reviewers hold a different opinion.

**Review:**

The paper titled 'Learning Hybrid Actor-Critic Maps for 6D Non-Prehensile Manipulation' presents a novel reinforcement learning approach for achieving 6D non-prehensile manipulation of objects by leveraging point cloud observations. The results of the study are highly promising, demonstrating an impressive 89% success rate on previously unseen objects and a noteworthy 50% success rate in real-world experiments using zero-shot transfer. Overall, the paper is well-written and effectively conveys its contributions.

While the hybrid actor-critic method is not entirely new to the reinforcement learning community, this work delivers a comprehensive and thoroughly validated package of their approach, which serves as a valuable reference for the entire community. I have thoroughly reviewed all the equations presented in the paper, and they demonstrate a high level of technical soundness.

**Quality Of The Limitations Section:**

Additional details required

**Questions For Rebuttal:**

I do not really have questions for this paper. However, please write more insights how can you tackle the problem of only 50% success rate in real-world experiment?

**Robotics Focus:**

Sufficient demonstration on hardware

**Summary Of Paper:**

The paper "Hybrid Actor-Critic Maps for Manipulation (HACMan): Learning 6D Non-Prehensile Manipulation using Point Cloud Observations" introduces a reinforcement learning approach for non-prehensile manipulation of objects.
HACMan proposes a novel action representation for non-prehensile manipulation, which is temporally-abstracted and spatially-grounded. The action representation consists of selecting a contact location from the object's point cloud and a set of motion parameters describing the robot's movements after making contact. The authors modify an existing off-policy reinforcement learning algorithm to learn in this hybrid discrete-continuous action representation.
Comparisons with alternative action representations show that HACMan outperforms the best baseline by achieving a success rate more than three times higher.

**Summary Of Recommendation:**

Having carefully reviewed this work, I am pleased to express my strong acceptance of this paper. The authors present a compelling and innovative reinforcement learning approach that addresses the challenging problem of achieving 6D non-prehensile manipulation using point cloud observations. Their results are truly remarkable, showcasing an impressive 89% success rate on unseen objects and a commendable 50% success rate in real-world experiments with zero-shot transfer.

The paper is well-written, exhibiting a clear and concise presentation of the research, methodologies, and findings. Moreover, the authors have provided a complete and rigorously verified package of their hybrid actor-critic method, making it an excellent reference for the reinforcement learning community.

---

### Official Review · Reviewer_Qqvw · 2023-07-22

**Confidence:** 4
**Originality:** Good
**Technical Quality:** Good
**Clarity Of Presentation:** Fair
**Impact:** 3

**Recommendation:**

Strong Accept: I recommend accepting the paper and will argue for my recommendation even if other reviewers hold a different opinion.

**Review:**

In general, the idea of directly defining the Q values on the surface of an object is interesting and promising. Also, the way the authors propose to train the respective models is sound.

My main issue with this paper is that the method could be explained more clearly.

The actor is defined as a function that takes the point cloud as input and outputs a set of actions. I assume this is realized with the per-point features of a pointnet++ style architecture? Similarly for the Q function.
This raises the question why the authors call the approach "hybrid". The fact that the method selects from a discrete set of surface points is an implementation detail and not fundamental to the problem. For example, one alternative way of implementing those mappings is to compute a global point cloud feature and then input a "query contact location". This still is using contact locations from obtained point cloud, but the resulting Q function/actor would also be able to interpolate between points from the point cloud. A baseline testing this would be highly appreciated.


"The goal flow is a 3D vector which is concatenated to the other features of the point cloud observation". I.e. each point in the point cloud is a 6 (or 7 dim?) vector? Maybe I missed it, what is the segmentation dim?


Small comments:
- Please consider alternatives to "hybrid". This is not a hybrid problem.
- Punctuation around equations could be improved (full stop after equation)
- "temporally-abstracted" and "spatially-grounded" are wrongly used in this context in my opinion.
- "our method does not rely on quasi-static assumptions" In my understanding, the method only takes as input observations of a single time slice, hence it by definition cannot properly reason about processes that involve inertia or second order dynamics. I agree that no explicit quasi-static assumptions are made in the method.

Relevant other papers:
- "Neural Descriptor Fields: SE (3)-Equivariant Object Representations for Manipulation", A. Simeonov et al.
- "Deep visual constraints: Neural implicit models for manipulation planning from visual input", J-S. Ha et al.

**Quality Of The Limitations Section:**

Limitations are addressed clearly

**Questions For Rebuttal:**

See review above.

**Robotics Focus:**

Sufficient demonstration on hardware

**Summary Of Paper:**

This paper addresses non-prehensile object manipulation within an RL framework. The authors propose to define the input observation space of an actor-critic RL method to be surface points obtained from a point cloud observations. This way, both the Q function and the actor network are defined on the surface of an object. At inference time, the method selects both a contact location on the object surface and an action applied at this point to the object. Results in simulation and with a real robot indicate that defining the actor and Q function networks on the surface of objects is advantageous compared to multiple baselines.

**Summary Of Recommendation:**

As written in the review above, I encourage the authors to improve the presentation of the paper to clarify some parts of the method section.
I am happy to raise my score if my points are addressed.

Post rebuttal: raised score to strong accept

---

### Decision · Program_Chairs · 2023-08-30

**Decision:**

Accept (Oral)

**Comment:**

Overall, the paper is well-written, receives three Strong Accept and one Weak Accept reviews. The rebuttal has addressed major concerns and issues. Please revise the paper in line with the rebuttal discussion for the camera-ready submission.

**Strengths**:

- The paper's writing is clear and effective.

- The core idea of directly defining Q values on contact points holds interest and promise.

- The combination of this novel action representation with the RL method may extend its influence beyond non-prehensile manipulation.

- The experiments are well-designed and the results are impressive, such as the generalization capability to unseen objects.

- The proposed critic map representation showcases an ability to handle multi-modal distributions.

**Weaknesses**:

- The central notion of defining Q values on an object's surface is not novel, has precedent in related literature, such as Contact-Graspnet for grasping, and the paper [20].

- Further clarity is needed to discern the primary contributors to the strong generalization performance with unseen objects.

- As outlined in the limitation section, the approach's reliance on depth measurements and the assumption of precise goal point tracking may restrict its applicability in real-world robotic scenarios.